# Enhanced Neural Empathic Responses in Patients with Spino-Bulbar Muscular Atrophy: An Electrophysiological Study

**DOI:** 10.3390/brainsci11010016

**Published:** 2020-12-24

**Authors:** Arianna Palmieri, Federica Meconi, Antonino Vallesi, Mariagrazia Capizzi, Emanuele Pick, Sonia Marcato, Johann R. Kleinbub, Gianni Sorarù, Paola Sessa

**Affiliations:** 1Department of Philosophy, Sociology, Education and Applied Psychology (FISPPA), University of Padova, Piazza Capitaniato, 35139 Padova, Italy; arianna.palmieri@unipd.it (A.P.); emanuele.pick@phd.unipd.it (E.P.); sonia.marcato87@gmail.com (S.M.); johann.kleinbub@gmail.com (J.R.K.); 2Padova Neuroscience Centre (PNC), University of Padova, Via Giuseppe Orus, 35131 Padova, Italy; 3School of Psychology, University of Birmingham, Edgbaston, Birmingham B15 2TT, UK; f.meconi.1@bham.ac.uk; 4Department of Neurosciences (DNS) & Padova Neuroscience Centre, University of Padova, Via Giustiniani, 35128 Padova, Italy; antonino.vallesi@unipd.it (A.V.); gianni.soraru@unipd.it (G.S.); 5Brain Imaging & Neural Dynamics Research Group, IRCCS San Camillo Hospital, 30126 Venice, Italy; 6EA 4556 Epsylon, Université Paul Valéry, 34090 Montpellier, France; mgcapizzi@hotmail.com; 7Department of Developmental and Socialization Psychology (DPSS), University of Padova, Via Venezia, 35131 Padova, Italy

**Keywords:** empathy, experience-sharing, mentalizing, event-related potentials, spino-bulbar muscular atrophy (SBMA)

## Abstract

Background: Spino-bulbar muscular atrophy is a rare genetic X-linked disease caused by testosterone insensitivity. An inverse correlation has been described between testosterone levels and empathic responses. The present study explored the profile of neural empathic responding in spino-bulbar muscular atrophy patients. Methods: Eighteen patients with spino-bulbar muscular atrophy and eighteen healthy male controls were enrolled in the study. Their event-related potentials were recorded during an “Empathy Task” designed to distinguish neural responses linked with experience-sharing (early response) and mentalizing (late response) components of empathy. The task involved the presentation of contextual information (painful vs. neutral sentences) and facial expressions (painful vs. neutral). An explicit dispositional empathy-related questionnaire was also administered to all participants, who were screened via neuropsychological battery tests that did not reveal potential cognitive deficits. Due to electrophysiological artefacts, data from 12 patients and 17 controls were finally included in the analyses. Results: Although patients and controls did not differ in terms of dispositional, explicit empathic self-ratings, notably conservative event-related potentials analyses (i.e., spatio-temporal permutation cluster analyses) showed a significantly greater experience-sharing neural response in patients compared to healthy controls in the Empathy-task when both contextual information and facial expressions were painful. Conclusion: The present study contributes to the characterization of the psychological profile of patients with spino-bulbar muscular atrophy, highlighting the peculiarities in enhanced neural responses underlying empathic reactions.

## 1. Introduction

Spino-bulbar muscular atrophy (SBMA), also known as Kennedy’s disease [1], is a rare adult-onset neuronopathy that belongs to the family of motor neuron diseases; symptoms usually emerge between 30 and 50 years of age [2]. Because of its X-linked nature, SBMA affects only the male biological sex, while females can be healthy carriers. Namely, SBMA is caused by an abnormal expansion of the glutamine’s trinucleotide CAG repeat length in the gene encoding the androgen receptors (ARs), thus, leading to insensitivity to androgens—mainly in terms of insensitivity to testosterone, the principal androgenic hormone.

Regardless of the individual variability in the effect of androgenic hormones concerning social status, as highlighted by van Honk et al. [3] and Eisenegger et al. [4], testosterone has been classically described to be inversely related to emotional behaviour, mainly to empathy [5,6]; for instance, females have a stronger attitude in empathizing than males. A single administration of testosterone in women leads to impairments in the ability to infer emotions, intentions, and the mental states of others. Thus, the administration of testosterone in women incurs an impairment in empathizing [6,7,8,9,10]. Artificially raised testosterone levels in adult males further inhibit empathic and altruistic behaviours [11]. These intriguing sex-related perspectives are well described by Baron–Cohen’s [7] Empathizing-Systemizing theory.

Typically, because of mutant ARs accumulation in SBMA patients’ motor neurons, a neurodegeneration in the spinal cord and brainstem occurs, incurring more evident SBMA symptoms, such as muscle weakness, cramps, fasciculation or twitching, tremor, dysarthria, and dysphagia. As body tissue cells of these patients uptake a reduced amount of testosterone, endocrinological alterations also result in gynecomastia, hypogonadism, and sometimes impotence [12,13]. Though less commonly reported than in other tissues, mutant ARs accumulation has been observed in SBMA patients’ brain tissues as well, such as in the frontal cortical areas, amygdala, and cingulate cortex [14,15,16].

Moreover, it is worth noting that a clinical impression often reported both by physicians and psychologists who have treated patients with SBMA is that these patients show peculiar behavioural patterns, such as extreme sensitivity, generosity, and cooperative or empathic behaviour, as already reported in the literature in such patients and argued to be compatible with testosterone deficiency [17,18]. In patients affected by Morris syndrome [19], a much more severe androgen insensitivity syndrome, the body tissues’ exposure to low levels of testosterone since infancy results in karyotypic (i.e., XX chromosomes) feminine behaviours [20,21]. Therefore, as a consequence of mutant ARs in the brain tissue, the brain and mind of patients with SBMA might express a sort of mild “feminisation” in some higher-level functions, such as empathic responses.

Despite the above-mentioned brain involvement, scarce and inconsistent literature has focused on the occurrence of cognitive impairment in SBMA. Although the presence of subtle cognitive deficits has been observed mainly, but not exclusively, in functions related to the fronto-temporal lobes [22,23,24,25], cognitive performance within normal range has also been described [26,27]. In this vein, in the author’s previous study focussing on language, executive and memory functions in a considerable number of patients [27], no impaired performance was observed. Intriguingly, these patients also showed superior performance compared to controls in the Prose Memory Test, a task aimed at assessing verbal memory recall of a narration of endangered people characterized by emotional content. Because no greater performance in any of the other memory tests administered to the patients was reported, the role of the Prose Memory Test’s emotional content could have driven the superior performance observed in these patients.

In light of such indications and considering the altered uptake of testosterone, which is inversely related to empathic behaviours in turn, it is surprising that empathic functioning has never been directly investigated. The aim of the present study was to investigate empathy functioning, in terms of neural responses that have been previously extensively linked with empathic reactions [28], in patients with SBMA. To fulfil this purpose, this study has adopted a paradigm developed by Sessa et al. [29], the “Empathy Task”, which provides direct evidence for temporally and functionally separable event-related potential (ERP) time windows linked to different components of empathy, namely experience-sharing and mentalizing. Of note, experience-sharing refers to affective empathy—that is, emotional resonance and the ability to share another person’s feelings—while mentalizing, which refers to cognitive empathy, involves the accurate recognition and understanding of another person’s thoughts and feelings [28]. Several findings have provided evidence that these two components of empathy (experience-sharing and mentalizing) are functionally dissociable and subserved by different neural systems [29,30,31,32,33,34].

Previous studies, using the Empathy Task [29], have shown an additive effect in non-clinical participant responses. Namely, painful faces, compared to neutral faces, have elicited the experience-sharing component of empathy, as reflected by the positive shift of only the relatively early P2 and N2–N3 components time-locked to face onsets. On the other hand, painful contexts elicited the mentalizing component of empathy compared to neutral contexts, as reflected by the positive shift of only the relatively later P3 component time-locked to face-onsets (see also other works from one of the authors and other groups in this field: [35,36,37,38,39,40,41,42]).

In line with these premises, we hypothesized that patients with SBMA, when compared to the control group, would present with enhanced neural empathic reactions, either both concerning the experience-sharing, detectable as increased early neural reactions (i.e., a larger positive shift in patients vs. controls in an early temporal window for painful conditions when compared to neutral conditions) and enhanced mentalizing, detectable as late enhanced neural reactions (i.e., similar positive shift in patients vs. controls in a later temporal window for painful conditions when compared to neutral conditions).

It is important to underline that this paradigm is not aimed at monitoring explicit measures of empathy (a construct that is difficult to capture in behavioural, self-reported terms [43]) but rather the implicit responses at the neural level that have been previously identified in many studies as being associated with empathy, as reported above.

Finally, explicit, dispositional empathy scores were also collected with the Interpersonal Reactivity Index (IRI; [44]), a self-report measure that was used to self-evaluate differences between patients and controls to eventually confirm the idea of enhanced empathy at an explicit, conscious level. Specifically, the affective and cognitive components of empathy were self-rated evaluated by the IRI subscales of Empathic Concern and Perspective Taking, respectively, as they are considered the gold standard in discriminating the different empathic dimensions. Although psychophysiological measures have already provided more effectiveness in objectively detecting empathic and emotional dispositional attitude [43,45], it was expected to find a likely increase in empathy scores detected by the IRI in patients with SBMA relative to controls, coherently to the expected enhancement of empathy-related physiological indices, i.e., the main hypothesis of the present contribution.

To rule out the possibility that cognitive deficits would have influenced the results, patients and controls underwent a screening battery of neuropsychological tests.

Previous literature findings informed this research questions concerning increased empathy in patients with SBMA, a hypothesis based on their AR insensitivity. Participants’ ERP responses were therefore considered the main dependent variables of the present investigation. Exploring empathic abilities in SBMA disease will aid researchers and clinicians to better delineate their peculiar, under-understood psychological profile and, in parallel, could represent a useful model to deepen the link between empathic functioning and testosterone uptake.

## 2. Materials and Methods

### 2.1. Study Design

During patients’ routine neurological check-up, a neurologist informed each patient about the research. Upon patients’ agreement, they arranged a date for the experimental session. Control participants were recruited from local volunteer associations. In this non-randomized case-control study, both patients and controls underwent a neuropsychological battery screening, lasting about 50 min, and the self-rated explicit dispositional empathy questionnaire, lasting about 10 min (see details in Section 2.3). Participants’ electrophysiological signals were recorded during the subsequent Empathy Task based on the orthogonal manipulation of facial expression (neutral vs. painful) and of context (written sentences with neutral vs. painful content preceding the facial expressions; see details in Section 2.4, Section 2.5 and Section 2.6). Behavioural responses in the Empathy Task were also collected. The entire procedure lasted about two hours and took place at the University Hospital of Padova. Written informed consent was obtained from all participants; the study was approved by the Ethical Committee of the University of Padova, protocol number 1561, and conducted in accordance with the principles of the Helsinki Declaration, as revised in 2013.

### 2.2. Participants

Patients were consecutively recruited among the outpatients attending the Motor Neuron Disease (MND) Centre of the Department of Neuroscience of the University of Padova in 2016–2017. They were all genetically diagnosed with SBMA and underwent neurological examination and neuropsychological evaluation by neuropsychologists trained in motor neuron disease management. The exclusion criteria were comorbidity with other neurological diseases, previous psychiatric diagnoses, current high use of psychoactive drugs, and not normal or corrected-to-normal vision. Among 20 consecutively attending patients, 18 were eligible according to the exclusion criteria, and all agreed to participate in the study. Eighteen control participants were recruited for whom the same exclusion criteria were applied.

The samples for the analyses concerning the neuropsychological battery screening and the explicit dispositional empathy questionnaires comprised all participants in each group, namely 18 patients and 18 controls. Six patients and one control were discarded from ERP analyses due to excessive electrophysiological artefacts, thus leaving 12 patients and 17 controls for ERP analyses. Participants demographic characteristics are present in Table 1.

### 2.3. Neuropsychological Screening and Explicit Dispositional Empathy Assessment

The neuropsychological screening assessment was composed of traditional neuropsychological tests. In detail, the Forward Digit Span [48], Backwards Digit Span [48], Prose Memory Test [49], Trail making test A and B [50], Phonemic Fluency Test [51], and Mental Rotation Test [52] were administered. No adjustments were made in the time-dependent tests based on verbal speech because none of the patients showed evidence of bulbar disabilities that might have affected their neuropsychological performance.

Explicit dispositional empathy of all participants was also collected by means of the Interpersonal Reactivity Index (IRI; [44]; [53] for the Italian validation). This self-report questionnaire is composed of 28 items subdivided into four subscales. In the present investigation, only the Empathic Concern and the Perspective Taking subscales were used, which align to the affective and cognitive components of empathy, respectively.

### 2.4. Empathy Task: Stimuli

The task and the stimuli were the same as those used in the author’s previous study (Figure 1; [29]). The stimuli included 32 Caucasian faces of 8 females and 8 males with either a neutral or painful expressions [36], and 32 sentences describing either a neutral or a painful context (e.g., “This person owns a yellow car” for the neutral context; “This person got a finger hammered” for the painful context). The stimuli were presented at the centre of the screen on a 19” LCD monitor controlled by a computer running E-prime 2 software [54] from a viewing distance of approximately 70 cm.

### 2.5. Empathy Task: Procedure

Each trial began with the presentation of a fixation cross at the centre of the screen (600 ms), followed by one of the sentences (i.e., a contextual cue; 3000 ms) describing either a neutral or a painful context. After a blank interval (800–1600 ms, jittered in steps of 100 ms), one of the faces (i.e., a perceptual cue) with either a neutral or a painful expression was displayed for 250 ms. A total of four possible conditions of painful and neutral cues for both sentences and faces were present, with each combination that was randomly picked at each trial. A painful condition was characterized by either a painful context, a painful facial expression or by both cues being painful. A neutral condition was characterized by both cues being neutral. In each of the three blocks, the full combination of the painful and neutral conditions was administered for a total of 16 trials per condition in each block.

Participants were first presented with a sentence describing either a painful or neutral context followed by a face with either a painful or neutral expression. They were required to carefully read the sentence and to decide whether the face had a painful or a neutral expression by pressing ‘J’ or ‘F’ on the keyboard. The mapping of response keys was counterbalanced across participants. At the end of each trial, and once the decision about the facial expression was made, they had to rate how much empathy they perceived for the depicted individual imagined in the preceding context on a 7-point Likert scale. Following a brief session of practice aimed at familiarizing participants with the task, they performed 192 trials (i.e., 64 for each block).

### 2.6. Empathy Task: Recording, Pre-Processing, and Analyses Designs

Electroencephalographic (EEG) activity was recorded using 64 electrodes distributed over the scalp in accordance with the international 10/20 system and placed on an elastic Easy-Cap (Herrsching, Germany) referenced to FCz. The EEG was re-referenced offline to the average of the left and right mastoids. The vertical electrooculogram (VEOG) was recorded from Fp1 and one external electrode placed below the left eye. The horizontal EOG was monitored through the scalp electrodes positioned in proximity to both eyes. The electrode impedance was kept below 10 kΩ.

EEG and EOG signals were amplified and digitized at a sampling rate of 250 Hz and filtered (pass band 0.01–80 Hz) during recording. An offline pass-band filter (0.01–30 Hz 24 dB/Octave) and a notch filter for the line noise (50 Hz) were applied before segmentation. Independent Component Analysis (ICA) was conducted to correct for blinks and eye movements using the semi-automated procedure implemented in BrainVision Analyzer Vers. 2.0.3 [55]; an average of 3.54 components per participant were removed (*SD* = 1.04, range 2–5). Trials associated with incorrect responses or contaminated by artefacts exceeding ± 120 μV on any channel were excluded from further analyses. This procedure resulted in patients providing a range of 15 to 43 analysed trials (neutral face and neutral context: *M*_1_ = 30.08, *SD*_1_ = 7.7; neutral face and painful context: *M*_2_ = 31.58, *SD*_2_ = 9.22; painful face and neutral context: *M*_3_ = 28.25, *SD*_3_ = 9.36; painful face and painful context: *M*_4_ = 29.58, *SD*_4_ = 7.54), whereas for the controls, a range of 17 to 46 trials were kept for the analysis (neutral face and neutral context: *M*_1_ = 36.47, *SD*_1_ = 6.04; neutral face and painful context: *M*_2_ = 36.41 *SD*_2_ = 6.10; painful face and neutral context: *M*_3_ = 36.18, *SD*_3_ = 7.32; painful face and painful context: *M*_4_ = 35.12, *SD*_4_ = 8.17). The number of trials kept for the patients was, on average, significantly lower than that for controls [*t*(27) = −2.29, *p* = 0.03]. The EEG was then segmented into 1200-ms epochs, starting at 200 ms prior to the onset of the faces. The epochs were baseline-corrected based on the mean activity during the 200 ms pre-stimulus period for each electrode site. Separate average waveforms for each condition were then generated time-locked to the presentation of the face stimuli as a function of the preceding context. Time-windows were chosen following the previous studies to investigate an earlier response (0–350 ms) for the experience-sharing responses and a late time-window (350–700 ms) for the mentalizing responses [29,41,42].

### 2.7. Statistical Analyses

#### 2.7.1. Demographic Characteristics, Neuropsychological Screening, and Explicit Dispositional Empathy Assessments

Due to the small sample size, data were analysed via a distribution-free test, the Mann–Whitney U test. Analyses were carried out with the R statistic software (Vers. 3.4.4, [56]). *p* values were corrected for multiple comparisons with the false discovery rate approach [57] using the R software package (Vers. 3.2.6). Only corrected *p* values were reported. The alpha level was set to 0.05 for this and all the subsequent analyses. In addition, for neuropsychological screening, and explicit dispositional empathy assessments, Cohen’s *d* was computed and reported.

#### 2.7.2. Behavioural Responses in the Empathy Task

Trials associated with response times (RT) exceeding each individual mean RT in a given condition +/− 2.5 *SD* and incorrect responses were excluded from the analyses (a total of 4.99% of discarded responses). Individual mean proportions of correct responses, RTs associated with correct responses and ratings were submitted to separate mixed ANOVAs, both considering facial expression (painful vs. neutral) and context (painful vs. neutral) as within-subject factors and the group (patients vs. controls) as a between-subject factors.

#### 2.7.3. ERP Responses in the Empathy Task

Given the relatively small sample size (i.e., 12 patients and 17 controls), justified by the rarity of SBMA, a nonparametric permutation approach of the EEG signal with spatio-temporal clustering (e.g., [58]) was chosen in the present study. An early (0–350 ms) and a late (350–700 ms) temporal window were considered, per the conspicuous evidence in favour of an early experience-sharing response and a late mentalizing response [29,41].

Cluster analyses with 1000 Monte Carlo random permutations were carried out unrestrictedly over all electrodes in the averaged time for each time-window and in comparing the three conditions where one or two cues were painful with the neutral condition, that is, when both cues were neutral. This was done separately for the patients and controls. Furthermore, to test the statistical difference between the groups concerning the empathic responses, differential responses of the relevant effects (i.e., ERPs for a painful face and neutral context minus the neutral condition; ERPs for a painful context and neutral faces minus the neutral condition; ERPs for both cues painful and the neutral condition) were computed and then tested between groups with the same cluster analysis procedure as with the independent samples.

## 3. Results

### 3.1. Demographic Characteristics

The samples for the analyses concerning the neuropsychological battery screening and the explicit dispositional empathy questionnaires were paired for age (patients: *M* = 56.11 years, *SD* = 9.93; controls: *M* = 56.78 years, *SD* = 10.75; *U* = 177.5, *p*_corr_ = 0.82) and education (patients: *M* = 13.36 years, *SD* = 3.30; controls: *M* = 14.92 years, *SD* = 3.23; *U* = 191.0, *p*_corr_ = 0.70). The subsamples used for ERP analyses were paired for age (patients: *M* = 52.67 years, *SD* = 9.15; controls: *M* = 56.24 years, *SD* = 10.82; *U* = 128.5, *p*_corr_ = 0.70) and education (patients: *M* = 13.50 years, *SD* = 4.21; controls: *M* = 13.76 years, *SD* = 3.05; *U* = 102.5, *p*_corr_ > 0.99).

### 3.2. Neuropsychological Screening Assessment

The group comparison in the neuropsychological screening assessment did not show significant differences in any test across the groups. In the Forward Digit Span (*U* = 172, *p*_corr_ = 0.70), the patients’ scores (*M* = 5.78, *SD* = 1.35) were smaller than the controls’ scores (*M =* 5.89, *SD* = 0.76) with a small effect size (*d* = 0.10). For the Backwards Digit Span assessment (*U* = 170, *p*_corr_ = 0.70), the patients’ scores (*M* = 4.50, *SD* = 1.10) were smaller than the controls’ (*M =* 4.56, *SD* = 0.92) with a very small effect size (*d* = 0.06). Next, in the Trail Making Test B–A (*U* = 134.5, *p*_corr_ = 0.95), the patients’ scores (*M* = 39.67, *SD* = 18.29) were higher than the controls’ scores (*M =* 34.94, *SD* = 10.78) with a small effect size (*d* = 0.31). For the Phonemic Fluency Test (*U* = 210, *p*_corr_ = 0.70), patients’ scores (*M* = 40.39, *SD* = 11.90) were smaller than the controls’ (*M =* 46.67, *SD* = 13.42) with a medium effect size (*d* = 0.50). Continuing, in the Prose Memory Test (*U* = 91.50, *p*_corr_ = 0.99), patients’ scores (*M* = 14.84, *SD* = 1.71) were greater than the controls’ scores (*M =* 13.92, *SD* = 1.42) with a medium effect size (*d* = 0.59). Finally, in the Mental Rotation Test (U = 155.50, *p*_corr_ = 0.88) patients’ scores (*M* = 6.72, *SD* = 4.90) were greater than the controls’ scores (*M =* 5.56, *SD* = 2.23) with a small effect size (*d* = 0.31). The tabular view of the results can be seen Appendix A.

### 3.3. Explicit Dispositional Empathy Assessment

The group comparison in the explicit dispositional empathy assessment did not show significant differences across groups, although, a trend in the expected direction was observed for the main affective subscales of the IRI Empathic Concern subscale (*U* = 115, *p*_corr_ = 0.70), in which the patients’ scores (*M* = 26.63, *SD* = 4.51) were greater than controls’ scores (*M =* 26.22, *SD* = 2.65) with a very small effect size (*d* = 0.11). The Perspective Taking subscale (*U* = 174.50, *p*_corr_ = 0.70) patients’ scores (*M* = 22.83, *SD* = 3.91) were smaller than that of the controls (*M =* 23.00, *SD* = 3.77) with a very small effect size (*d* = 0.04). The tabular view of the results can be seen in Appendix A.

### 3.4. Behavioural Responses in the Empathy Task

The ANOVA conducted for the mean proportion of correct responses showed a significant interaction between facial expression and context (*F*(1, 27) = 4.451, *p* = 0.044, *MS*_e_ = 0.001, *η*_p_^2^ = 0.142) in the direction of a better performance for neutral (*M* = 0.983, *SE* = 0.005) compared to painful faces preceded by a neutral context (*M* = 0.969, *SE* = 0.008) and similar performance for neutral (*M* = 0.974, *SE* = 0.009) and painful faces (*M* = 0.979, *SE* = 0.006) preceded by a painful context. Bonferroni corrected planned comparisons revealed no significant contrast (max *t*(27) = 1.520, min *p*_corr_ = 0.140). None of the main effects were significant (all *F*s < 1). The interaction between the within-subject factors and the group did not reach the significance level (*F*(1, 27) = 1.650, *p* = 0.210, *η*_p_^2^ = 0.058). Furthermore, the RTs did not show any significant effect (max *F*(1, 27) = 2.474; min *p* = 0.127, max *η*_p_^2^ = 0.084).

The ANOVA conducted for the ratings showed a main effect for facial expression (*F*(1, 27) = 8.308, *p* = 0.008, *MS*_e_ = 2.364, *η*_p_^2^ = 0.235), context (*F*(1, 27) = 16.650, *p* = 0.000358, *MS*_e_ = 1.166, *η*_p_^2^ = 0.381), and the interaction between these factors (*F*(1, 27) = 16.215, *p* = 0.000412, *MS*_e_ = 1.880, *η*_p_^2^ = 0.375). Bonferroni corrected planned comparisons revealed that participants rated their perceived empathy as higher when painful faces were preceded by a painful context compared to a neutral one (*M*_diff_ = −1.872 (−2.627, −1.116); *t*(27) = 5.085, *p*_corr_ = 0.000024). Painful faces were rated as eliciting more empathy than neutral faces when preceded by a painful context (*M*_diff_ = −1.876 (−2.705, −1.048); *t*(27) = 4.65, *p*_corr_ = 0.000079). Figure 2 shows a bar-graph of the mean rating scores for each condition with standard errors and significant Bonferroni corrected comparisons. The interaction with the group was not significant (*F* < 1).

### 3.5. ERP Responses in the Empathy Task

Figure 3 shows the grand averages of the ERP components time-locked to the onset of the face as a function of the context recorded at the sites that formed a cluster. Each experimental condition is presented superimposed with the ERPs elicited in the neutral condition (i.e., neutral context and neutral facial expression) from the left to the right (i.e., left panel: neutral context and painful facial expression; middle panel: painful context and neutral facial expression; right panel: painful context and painful facial expression). The two topographies for each graph represent the scalp activity in the early and late time-windows. Panels A, B, and C show results from the patients with SBMA, panels D, E, and F show the results from the controls and panels G, H, and I show the differential ERPs computed as the difference of each experimental condition minus the neutral condition for the SBMA patients and the controls.

#### 3.5.1. Patients

The cluster-analysis carried out in the first time-window (0–350 ms) revealed that painful faces preceded by painful contexts elicited significantly more positive ERPs compared to the neutral condition (*p*_corr_ = 0.00099), that is, when both cues were neutral. The same comparison was not significant in the second time window (350–700 ms, *p*_corr_ = 0.12). Similarly, in the first time-window (but not in the second time-window; *p*_corr_ = 0.08), painful faces preceded by a neutral context produced significantly more positive ERPs compared to the neutral condition (*p*_corr_ = 0.003). By contrast, neutral faces preceded by painful contexts elicited significantly more positive ERPs compared to the neutral condition in the later time window (350–700 ms, *p*_corr_ = 0.019) but not in the first time-window (*p*_corr_ = 0.49).

#### 3.5.2. Control Group

The cluster-analysis carried out in both time windows showed a significant cluster for painful faces preceded by painful contexts compared to the neutral condition in the early time-window (0–350 ms), indicating an early empathic reaction to physical pain in these participants only when both cues were painful (*p*_corr_ = 0.033). No other condition showed any significant cluster.

#### 3.5.3. Patients vs. Controls

The direct comparisons of the effects between patients and controls showed a significant cluster only when both cues were painful in the first time-window linked with the experience-sharing component of empathy (0–350 ms, *p*_corr_ = 0.019) but not for the other contrasts (min *p*_corr_ = 0.079 for the effect in which the face was painful and the context was neutral), including those related to the mentalizing component of empathy. Figure 3 depicts the results of the cluster-analysis. The representation of the results follows Rousselet et al. [59].

## 4. Discussion

The findings of the present study revealed enhanced neural empathic responses in patients with X-linked SBMA disease. Namely, because the disease implies mutations at the AR level and as, in turn, testosterone is associated with an empathic attitude, it was initially hypothesized that these patients would show a different pattern of neural empathic responses compared to a group of healthy males, comparable in neuropsychological performance, as controls. Greater empathic responses in patients in terms of (1) central nervous system activation and, possibly, (2) explicit self-rated dispositional style were expected.

For the first investigative purpose, the Empathy Task developed by Sessa et al. [29] was used, to examine the two components of empathy in these patients compared to healthy controls: the affective one (i.e., experience-sharing) and the cognitive one (i.e., mentalizing). Consistent with the main hypothesis, the early empathic response (experience-sharing, i.e., the more affective component of empathy) was significantly larger in patients when compared to controls, especially when both empathy cues (i.e., context and face) were painful. It is well known that empathy is considered to be an adaptive and flexible process, and sensitive to information of a contextual nature (such as affective cues, threatening information, group membership; see [60]). From this perspective, the result of a greater empathic neural response in patients compared to controls in the experience-sharing window when both empathic cues were painful seems to indicate greater integrative capacity in patients than controls. Patients also showed an empathic response for the painful faces preceded by neutral contexts. This response was not observed in the controls, nor any significant difference between the effects in the two groups was observed.

These results should be interpreted with caution: on the one hand, a very large literature has supported the relationship between these neural responses and empathy (e.g., [28]); on the other hand, these increased neural responses do not directly inform about behavioural, self-reported responses of an empathic dispositional nature of participants. Nevertheless, it is interesting to note a trend toward an increased affective dispositional empathy in patients compared to controls, as evidenced by the Empathic Concern subscale of the IRI, which, however, does not reach statistical significance level.

In detail, despite the caveat in interpreting the results, it is worthy to note that the present findings are fully in line with a whole series of experimental evidence described below.

First, the greater neural reactivity identified in both experience-sharing and mentalizing aligns well with previous evidence about testosterone’s impact on empathic attitudes. Namely, as already highlighted in the introduction section, there is a great amount of research strongly suggesting that testosterone is inversely correlated with empathic skills [5,6,7,8,9,10]. To cite further notable examples, Zak et al. [11] showed that in the “Ultimate Game” (see [61]), men with artificially raised testosterone were significantly less prosocial compared to themselves on a placebo treatment. The authors concluded that elevated testosterone caused the men to display psychopathic traits. Furthermore, literature consistent with aforementioned data (e.g., [62]) has already identified strong connections between high testosterone levels in males and psychopathic traits and disorders, which, in turn, are mainly clinically characterized by aggressive behaviours and lack of empathy [63]. Analogously, patients suffering from an autism spectrum disorder, which is typically characterized by dramatically low levels of empathy, usually have greater levels of circulating testosterone compared to non-clinical people of the same gender and age [64]. Within this framework, the theory of “extreme male brain” in autism has been developed [7].

Second, from a broader perspective, the present findings are consistent with the idea of a more enhanced empathic attitude in feminine brains [65] and with Empathizing-Systemizing theory by Baron–Cohen [7,8], which supports evolutionary roots for the phenomenon of females displaying a stronger drive to empathize than males on average. The role of testosterone is at the basis of this empirically-validated theory, distinguishing males and females in their emotional processes and empathizing abilities in terms of the more prominent female attitude to empathize and share affectivity with others and the male attitude to logically analyse or identify the rules that govern a system in order to predict how it will behave. CAG triplet expansion in muscle, blood or salivary biopsy samples: on a speculative level, however, it is ontologically legitimate to associate the main abnormality of these male patients, i.e., insensitivity in testosterone uptake—a crucial feature that makes these patients different from healthy individuals and that generates the medical sequelae also related to body “feminisation” described in the introductory paragraph—with significantly different performance relative to the electroencephalographic pattern in response to tasks eliciting an empathic response.

Third, the findings of the present study, aligned with the hypothesis concerning enhanced affective empathy, might be supported from a neurobiological perspective as well. In neuroimaging studies, it has been demonstrated that the affective component of empathy engages the anterior cingulate cortex, premotor cortex, inferior parietal lobule and inferior frontal gyrus, anterior insula, and amygdala [28,32,33]. Additionally, the cognitive component of empathy engages the ventral-medial prefrontal cortex, precuneus, posterior cingulate cortex, temporal pole, and temporo-parietal junction [28,32,33]. The impairment of testosterone uptake in terms of mutant AR aggregation has also been described as occurring in these areas in patients (see Introduction; [14,15,16]); thus, it may have favoured the effect observed in the ERP empathic responses, as underpinned by the above-mentioned areas.

It should be mentioned that control participants displayed an unexpected result in the late time window, i.e., a diminished empathetic reaction for painful contexts. Differently from previous literature on empathy for physical pain and on pain recognition in healthy subjects (e.g., [29,66]), greater ERP empathic reactions in the time-window related to mentalizing were not found. One possible explanation for this result relies on conspicuous evidence highlighting the role of age and biological sex in empathy (for the age variable, see [67,68,69]; for the biological sex variable, see [70,71,72,73]). In the previous study of Sessa et al. [29], the sample was mainly composed of healthy female students at the University of Padova, with a mean age of 25 years. Some EEG and ERP studies have examined the possible differences in the empathic response in the two biological sexes; consistent reduced emotional and empathic responses for male individuals compared to female individuals were observed in the late positive potential (LPP) and in the P3 [35,74] as well as in the suppression of the mu rhythm during the observation of painful and non-painful situations (higher in female than in male participants; [75]). Additionally, Han et al. [35] used a similar paradigm to the one used in the present study and observed differences in P3 empathy-related responses (in a time window between 340 and 540 ms) between male and female participants, such that greater P3 responses to painful stimuli (vs. non-painful ones) were observed in females compared to males. In Coll’s [66] meta-analysis, which included 40 ERP studies on empathy for pain, the authors observed the modulation of the P3 component as a very reliable index of vicarious pain recognition. However, it is worth mentioning that in Coll’s meta-analysis, only cues with direct nociceptive stimulation (rather than painful facial expressions) were included. Future work should therefore assess the possibility that older male adults might show lower levels of cognitive empathy than expected.

As already briefly mentioned at the beginning of this section, contrary to the expectations of the secondary hypothesis, the explicit dispositional empathy assessment did not show statistically significant differences between the patients and controls. Namely, the patients’ scores obtained from the IRI Empathic Concern subscale [44], conceived as one of the most representative measurement of self-rated affective empathy, and the Perspective Taking subscale, conceived as one of the ideal measurements of cognitive empathy [76], did not reveal significantly higher scores in patients compared to control participants. However, it is interesting to note that the experience-sharing related subscale (Empathic Concern) results were higher in patients than in healthy controls. This aligns to what has been found in females compared to males since the first validation of the index in the 1970s by Davis [44].

The fact that in such a limited sample—dictated by the rarity of the disease—no differences were found between self-reported empathy data and empathy-related psychophysiological data should not, in our opinion, cast excessive doubt on the generalizability of the results, although consistent data would have been more straightforwardly interpretable. Generally speaking, a limitation of the small sample size is that it does not provide a definite comparison between the two groups in the self-report questionnaires. Furthermore, although there is a great amount of literature on empathy measured through self-report questionnaires, such methodology present severe intrinsic limits, such as presentation and response bias, subjectivity and susceptibility to motivational distortion, and social desirability [45,77]. For such reasons, although self-report questionnaires are an essential tool in empathy investigation from the current literature perspective, the main attention has been focused on psychophysiological measures that have already been proven more effective in objectively detecting the nuances of this empathy construct [43,45,77]. It is in fact common clinical experience that certain patients, such as those with narcissistic functioning, tend to report a much higher level of empathy aptitude than is actually experienced [78]. On the other hand, people with damaged self-esteem, such as those with vulnerability to depression, tend to underestimate their empathic abilities, as well as any other positive attitude attributable to themselves.

Of note, the enhanced empathic reactions found, characterizing the cognitive profile of patients with SBMA, did not permit concluding that this profile is strictly causally linked with impaired testosterone uptake. Nonetheless, in light of the evidence discussed above, the authors are confident that their interpretation is a sound working hypothesis that invites further investigations to explore more directly the possible causal relationship between testosterone uptake in these patients and their empathic abilities. Unfortunately, a direct measurement of the correlation between circulating testosterone as inferred by hematic or salivary sample from patients and ERP results would have been nonsense statistics because the impaired capacity of tissue to capture the testosterone results in a paradoxical high level of circulating testosterone, which is systematically degraded without body tissues’ metabolic assumption. On the other hand, to use CAG repeat length to infer the amount of mutant ARs would be useless as well; CAG repeat length varies according to body tissues, and such a difference is specific to each patient—the so-called phenomena of somatic mosaicism, common in several neurological disorders [16]. Thus, the CAG muscle biopsy repeat length, employed to ascertain a SBMA diagnosis, is useless for inferring the precise brain CAG repeat length that would be important to the eventual correlation with empathy response activation.

As a whole, these findings may shed new light on empathy functioning in patients with SBMA. Interestingly, other pathologies involving motor neurons’ degeneration, such as amyotrophic lateral sclerosis, present a different scenario; for instance, patients affected by such a disease show significantly impaired performance in mentalizing abilities, in affective empathic competencies and in social skills (for a review, see [79]). The intriguingly increased neural response found in patients with SBMA could be ascribed to the so-called “paradoxical functional facilitation” in neurological diseases [80] and could increase knowledge of the taxonomy of empathic functioning studied through the neurological disease perspective (for a review, see [81]).

## 5. Conclusions

Although patients with SBMA seem to have more enhanced empathic attitude—at least from their psychophysiological reactivity perspective—it is important to highlight the caveats of the present study. First, the sample size was relatively small and lacked a third female comparison group. Moreover, although the interpretation of the results preferred the hypothesis of the crucial role of impaired testosterone uptake in the theoretical premises and in the explanation of the results, the authors are aware that there are many other factors in addition to the normal demographic variables already considered in this study (age, gender, marital status, and educational level) that could have differed between the patients and controls. For example, although these patients did not reveal deflected mood, were not on psychotropic medication, and conducted a completely normal working and social life, the implications of the disease (e.g., mild gynecomastia and frequent infertility) may have led these participants to experience suffering and social inadequacy. This experience, in turn, may have favoured the development of a natural ability to tune in and better understand the pain of others—in other words, empathic attitude [82]. Furthermore, there is also evidence of a possible interaction between testosterone and cortisol in association with empathy (the “dual-hormone hypothesis”; [83]), suggesting that testosterone alone cannot entirely explain empathic abilities. For the present investigation, unfortunately, cortisol measurements for the patients and participants of the control group have not been collected, and it is hoped that this clarification might be useful for future studies in the field.

This study may be beneficial for both clinical and research perspectives, first, by contributing to the characterization of the psychological profile of patients with SBMA compared to other neurological diseases. As empathic functioning has important implications for social competency, understanding the mental peculiarities of patients can be useful, for instance, for doctor-patient relationship management. Second, from a theoretical point of view, SBMA can represent an interesting model to further reflect on the influence of sexual hormones on empathy.

## Figures and Tables

**Figure 1 brainsci-11-00016-f001:**
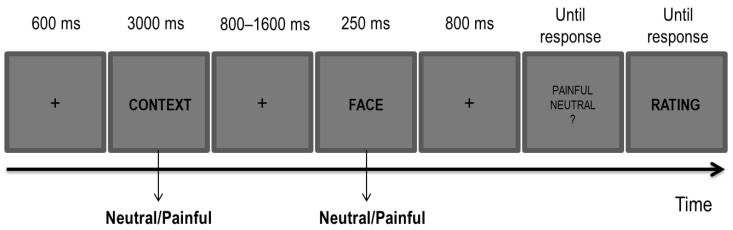
Experimental paradigm for the Empathy Task.

**Figure 2 brainsci-11-00016-f002:**
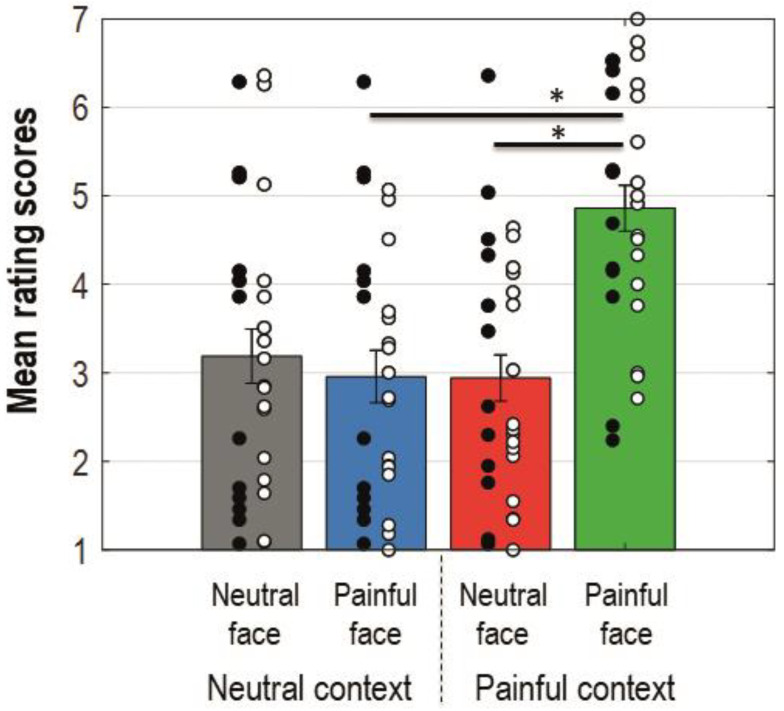
Bar-graphs displaying mean rating scores for each condition collapsed on group dimension. Error bars represent standard errors; black dots represent single patients; white dots represent single control participants. Colours of the bars match colours of the ERP waveforms for corresponding conditions. An asterisk indicates a statistically significant comparison.

**Figure 3 brainsci-11-00016-f003:**
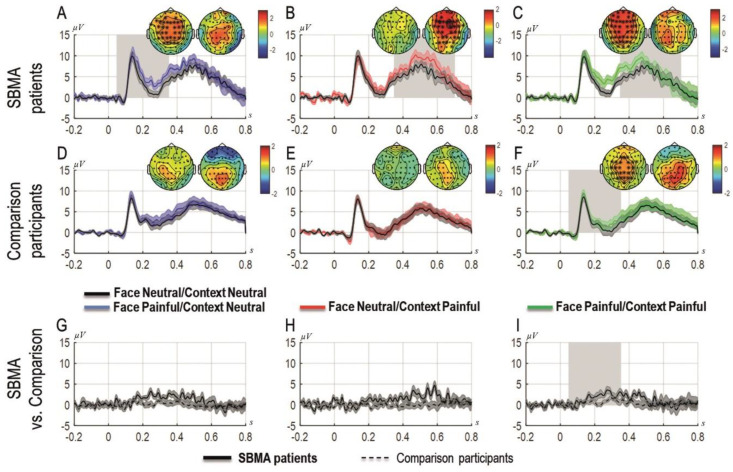
Grand-averages of ERPs at cluster sites for patients with SBMA and for controls separately and compared, for painful faces preceded by neutral contexts (**A**,**D**,**G**), for neutral faces preceded by painful contexts (**B**,**E**,**H**) and for painful faces preceded by painful contexts (**C**,**F**,**I**). Topographies are shown for each time window panel A to F (the left topography shows the scalp distribution of the activity in the 0–350 ms time-window; the right topography shows the scalp distribution of the activity in the 350–700 ms time-window). Asterisks represent the sites that form a cluster; the ERPs are plotted as the average of the activity in the significant cluster. The black dots represent the time-points with significant effects. Shades represent standard errors. Panels G, H, I represent differential responses between the conditions represented on the upper panels for patients with SBMA and controls.

**Table 1 brainsci-11-00016-t001:** Demographic characteristics of patients with SBMA and the control group.

ID	Age (Years)	Education (Years)	CAG Repeats	ADL Score	EEG Analyses
Patients					
P01	52	18	44	1	yes
P02	66	13	42	NA	no
P03	53	8	45	NA	yes
P04	62	8	45	2	no
P05	61	13	48	1	yes
P06	54	10	48	1	yes
P07	45	18	45	0	yes
P08	46	18	50	1	yes
P09	62	13	45	1	no
P10	58	13	44	1	yes
P11	52	13	46	1	yes
P12	40	16	45	NA	yes
P13	54	13	44	1	no
P14	59	13	44	2	yes
P15	62	13	43	2	yes
P16	74	8	45	2	no
P17	72	5	42	2	no
P18	38	17	48	0	yes
Controls					
C01	61	13			yes
C02	35	13			yes
C03	41	13			yes
C04	50	8			yes
C05	40	13			yes
C06	50	18			yes
C07	45	18			yes
C08	61	18			yes
C09	62	16			yes
C10	66	18			no
C11	64	13			yes
C12	64	18			yes
C13	58	13			yes
C14	66	13			yes
C15	70	8			yes
C16	68	13			yes
C17	66	13			yes
C18	55	13			yes

CAG repeats: number of CAG repeats, n > 37 is pathological [46]; Activity of daily living (ADL) score was assigned as follows: 0 = normal, 1 = mild weakness of limb muscles, climbs stairs easily but aware of weakness, 2 = mild to moderate weakness, climbs stairs with difficulty and generally uses a cane, and 3 = moderate to severe weakness, uses a wheelchair most of the time or mostly recumbent [47].

## Data Availability

All data supporting the results in this paper are available for download at the following address: http://doi.org/10.17605/OSF.IO/MSJ38. For other types of material, please contact the corresponding author.

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
