# Peer review of "Enhanced Neural Empathic Responses in Patients with Spino-Bulbar Muscular Atrophy: An Electrophysiological Study"

_brainsci, 2020, doi:10.3390/brainsci11010016_

Round 1
Reviewer 1 Report
This neurophysiological study on Spino-Bulbar Muscular Atrophy (SBMA) patients demonstrated enhancement only of one of the empathy components, i.e., experience-sharing ERP component in patients compared to healthy controls in the Empathy Task. This finding contributes to the characterization of the psychological profile of the SBMA patients and, more generally, may contribute to the neurophysiological underpinning of the effects the testosterone levels exert on empathic behavior.
This rather detailed study with an excellent discussion section could profit from repharasing/clarifications/editing of the following sentences:
- Line 88:
- Line 127 – too long, is this sentence really necessary in this form?
- Line 150: The samples?
- Line 209:
- Line 376: ''mentalizing identified aligns well’’?
- Line 382: ''Furthermore, consistent literature''?
- Lines 407 and 408: consider rephrasing and editing both sentences
- Line 412: how about ''In the previous study of Sessa and colleagues (28).....''
- Line 417: consider replacing ''and in the supression'' to ''as well as in the....''
- Line 420: check the articles
- Lines 438-441 ¸
- Line 469.....: clarify ''although we have priviledged the interpretative hypothesis.....''
- Line 499: ''architectural experimental design''? Would that be experimental paradigm design?
Minor comments:.
- Please, make sure that all the acronyms are defined on their first appearance only (e.g., line 20 and 41 for SBMA) .
- The contributions of the authors are specified in a separate section. There is no need to provide such information in the Method section (lines 125, 143, 145).
Author Response
Response to Reviewer 1 Comments
- Comments and Suggestions for Authors
- Line 127 – too long, is this sentence really necessary in this form?
Modified, thank you. Line 151
- Line 150: The samples?
Added numerosity of samples, thank you. Line 176-178
- Line 209:
Added citation, thank you. Line 238
- Line 376: ''mentalizing identified aligns well’’?
Rephrased, thank you. Line 422-423
- Line 382: ''Furthermore, consistent literature''?
Rephrased, thank you. Line 429
- Lines 407 and 408: consider rephrasing and editing both sentences
Rephrased, thank you. Line 460-462
- Line 412: how about ''In the previous study of Sessa and colleagues (28).....''
Rephrased, thank you. Line 466
- Line 417: consider replacing ''and in the supression'' to ''as well as in the....''
Done, thank you. Line 470
- Lines 438-441 ¸
Rephrased, thank you. Line 496-499
- Line 469.....: clarify ''although we have priviledged the interpretative hypothesis.....''
Rephrased, thank you. Line 534
- Line 499: ''architectural experimental design''? Would that be experimental paradigm design?
Rephrased, thank you
- Minor comments:.
- Please, make sure that all the acronyms are defined on their first appearance only (e.g., line 20 and 41 for SBMA) .
Done, thank you
- The contributions of the authors are specified in a separate section. There is no need to provide such information in the Method section(lines 125, 143, 145)
Done, thank you
We thank the Reviewer for his/her insightful comments, and we hope the Reviewer is satisfied with our changes to the text (in green in the manuscript).
Reviewer 2 Report
The manuscript with the title ‘Enhanced Neural Empathic Response in Patients with Spino-Bulbar Muscular Atrophy: An Electrophysiological Study’ authored by Palmeri et al. aims to study the involvement of testosterone in empathy. Their approach to investigate patients with SBMA is very interesting. However, the interpretation of their data is hard to follow. They claim-even in the title- that patients show enhanced empathy. Behavioral results, on the other hand, do not show significant differences between groups. As far as I understand, patients show brain responses (ERPs) to the stimuli that were not found in controls. Interpretation of this lack in response in controls as increased empathy in patients should be done very carefully and clearly marked as speculation. The direct comparison of patients and controls reveals differences only when the context was painful. What is the authors interpretation of this result? The over-interpretation of the results in addition to the limitations of the study already mentioned by the authors brings me to the conclusion that I recommend revision of the manuscript in its current state. Authors would have to make it more clear how their findings allow for their interpretation.
Minor comment:
Why was patient P02 matched with C02 and not C18 and P18 with P2. According to the demographic characteristics this match would have made more sense.
Author Response
Response to Reviewer 2 Comments
- Comments and Suggestions for Authors
- The manuscript with the title ‘Enhanced Neural Empathic Response in Patients with Spino-Bulbar Muscular Atrophy: An Electrophysiological Study’ authored by Palmeri et al. aims to study the involvement of testosterone in empathy. Their approach to investigate patients with SBMA is very interesting. However, the interpretation of their data is hard to follow. They claim-even in the title- that patients show enhanced empathy. Behavioral results, on the other hand, do not show significant differences between groups. As far as I understand, patients show brain responses (ERPs) to the stimuli that were not found in controls. Interpretation of this lack in response in controls as increased empathy in patients should be done very carefully and clearly marked as speculation. The direct comparison of patients and controls reveals differences only when the context was painful. What is the authors interpretation of this result? The over-interpretation of the results in addition to the limitations of the study already mentioned by the authors brings me to the conclusion that I recommend revision of the manuscript in its current state. Authors would have to make it more clear how their findings allow for their interpretation.
We thank the Reviewer for these important comments. Although the title states that the evidence reported in our work refers to neural responses previously (and widely) associated with empathy, the Reviewer is perfectly right in suggesting that in some paragraphs of the manuscript we have oversimplified our interpretation. With this in mind, we changed all the sentences in which it was stated that patients showed greater empathy than controls, clarifying that the results we observed indicate larger neural “reactions” in patients compared to controls in some experimental conditions and that these “reactions” have been associated with empathy in several previously published studies (as highlighted in both the introductory and discussion sections) by different research groups (lines 35, 38, 50, 101, 394, 397, 531). In particular, in the Discussion, we have clearly modified some critical sentences to clarify that some of our interpretations are speculative (albeit supported by some interesting experimental evidence) (lines: 414-421, 436, 491-494, 502-506, 522).
Furthermore, the Reviewer correctly observed that in the direct comparison between patients and control subjects, the significant difference refers to the condition in which both cues (capable of triggering an empathic response) were painful (lines: 404-406). In this regard, we clarified in the Discussion that this effect is not surprising in our opinion and is perfectly in line with all the experimental evidence that underlines how empathy is an extremely flexible and contextually situated response (also in neural terms) (lines: 443-449). In this regard we cite the interesting review by Melloni, Lopez & Ibanez (2014; CABN) and suggest that patients show a greater ability to integrate cues from different sources (in this case a verbal information and a facial expression) in an early processing window associated with experience-sharing (line 406-411).
- Why was patient P02 matched with C02 and not C18 and P18 with P2. According to the demographic characteristics this match would have made more sense.
Thank you, the Reviewer is correct. The order of the codes for patients and controls (P1, P2, C1, C2, etc.) has been given in the same order of their assessment. First we started collecting patients data, and only after we started collecting controls data. Therefore, the order of appearance in the table does not correspond to the one-to-one matching performed. To avoid such confusion, we reorganized the table in order to have one line for each participant. Moreover, none of the tables displayed (i.e., old and new version) affects the statistical analyses performed because the two groups were not different in their mean values of the demographic variables. We hope that this modification suits the expectation of the Reviewer and makes the table clearer. (line 180)
We hope the Reviewer is satisfied with our changes to the text (in light blue in the manuscript) and we thank the Reviewer again for these suggestions aimed at the caution in interpreting our results.
Reviewer 3 Report
I realize that a great work and time has been devoted to this paper. The Neural Empathic Responses in Patients with Spino–Bulbar Muscular Atrophy is of great significance, so I appreciate authors examining this topic.
The paper has a lot of strengths but I think that some changes should be recommended.
Abstract:
Readers should be able to read the abstract in isolation and understand what you have done, and its implications. So I recommend to the authors to expand the “Results” with more information, because it is unclear which correlations are significant.
I would suggest to the authors to avoid the word “we/our”, writing in impersonal mode as scientific style along the manuscript.
Please, avoid using abbreviations in the abstract.
Introduction:
Do the authors have an hypothesis? In this case, I suggest the authors to write correctly and well formulated the hypotheses. And please, do the same with the aim of the study.
Methodology:
It is unclear the design of study (case-control, cohorts, clinical trial).
Plese, specify if the assignment was randomized and if there was double blinding in each group.
Results and Discussion:
The results and discussion are good and exhaustive.
Please, include a section for conclusions.
I hope that these recommendations do not discourage the authors and I want to recommend the authors to continue working on this paper.
Author Response
Response to Reviewer 3 Comments
- Comments and Suggestions for Authors
- I realize that a great work and time has been devoted to this paper. The Neural Empathic Responses in Patients with Spino–Bulbar Muscular Atrophy is of great significance, so I appreciate authors examining this topic.
- The paper has a lot of strengths but I think that some changes should be recommended.
- Abstract:
- Readers should be able to read the abstract in isolation and understand what you have done, and its implications. So I recommend to the authors to expand the “Results” with more information, because it is unclear which correlations are significant.
We thank the Reviewer for this helpful suggestion. We have modified the abstract so that it can be more complete, especially in relation to the results. Line 33-50
- I would suggest to the authors to avoid the word “we/our”, writing in impersonal mode as scientific style along the manuscript.
Done, thank you.
- Please, avoid using abbreviations in the abstract.
Done, thank you.
- Introduction:
- Do the authors have an hypothesis? In this case, I suggest the authors to write correctly and well formulated the hypotheses. And please, do the same with the aim of the study.
Thank you for this important comment. We have modified one of the paragraph in the Introduction to better explain our hypothesis as follows: “In line with these premises, we hypothesized that patients with SBMA, when compared to the control group, would present with enhanced neural empathic reactions, either both concerning the experience-sharing, detectable as increased early neural reactions (i.e., a larger positive shift in patients vs. controls in an early temporal window for painful conditions when compared to neutral conditions) and/or enhanced mentalizing, detectable as late enhanced neural reactions (i.e., similar positive shift in patients vs. controls in a later temporal window for painful conditions when compared to neutral conditions).” Line 119-129, 135-139
- It is unclear the design of study (case-control, cohorts, clinical trial).
Clarified in the text, thank you. Line 153
- Plese, specify if the assignment was randomized and if there was double blinding in each group.
Clarified in the text, thank you. Line 153
- Please, include a section for conclusions.
The last part of the Discussion was already conclusive from a theoretical standpoint. Therefore, we separated it from the Discussion so to give autonomy to the Conclusions section. Thank you. Line 530
We hope the Reviewer is satisfied with our changes to the text (in yellow in the manuscript) and we thank the Reviewer again for these suggestions aimed at improving the quality of the manuscript.
Round 2
Reviewer 2 Report
The authors addressed all of my concerns.